# Changes in Preexisting Temporomandibular Joint Clicking after Orthognathic Surgery in Patients with Mandibular Prognathism

**DOI:** 10.3390/bioengineering9120725

**Published:** 2022-11-24

**Authors:** Chun-Ming Chen, Pei-Jung Chen, Han-Jen Hsu

**Affiliations:** 1School of Dentistry, College of Dental Medicine, Kaohsiung Medical University, Kaohsiung 807, Taiwan; 2Department of Oral and Maxillofacial Surgery, Kaohsiung Medical University, Kaohsiung 807, Taiwan; 3School of Oral Hygiene, College of Dental Science, Kaohsiung Medical University, Kaohsiung 807, Taiwan

**Keywords:** intraoral vertical ramus osteotomy, mandibular prognathism, orthognathic surgery, sagittal split ramus osteotomy, temporomandibular joint clicking

## Abstract

This study aimed to investigate the changes in preexisting temporomandibular joint (TMJ) clicking and the new incidence of TMJ clicking after orthognathic surgery. A total of 60 patients (30 men and 30 women) with mandibular prognathism underwent intraoral vertical ramus osteotomy (IVRO) for a mandibular setback. The setback amount and TMJ clicking symptoms (preoperative and one year postoperative) were recorded. To assess the risk of new incidence of TMJ clicking in asymptomatic patients, the cutoff value for postoperative mandibular setback was set at 8 mm. The left and right mandibular setbacks were 11.1 and 10.9 mm in men, respectively, and 10.7 and 10.0 mm in women, respectively. Thus, no difference in setback amount on either side was observed between the sexes. The improvement rate in patients with preexisting TMJ clicking was 69.2% (18 of 26 sides); the postoperative improvement rates were 71.4% (setback amount > 8 mm) and 60% (setback amount ≤ 8 mm). IVRO may reduce the severity of preexisting TMJ clicking. A high setback amount (>8 mm) may not be associated with a considerable increase in the risk of postoperative TMJ clicking.

## 1. Introduction

Each individual exhibits unique facial bone growth and development due to the congenital nature of tissue and organic system function. Therefore, changes in the pattern and rate of growth lead to abnormalities in facial bone structure, resulting in varying degrees of jaw deformity and malocclusion [1]. Mandibular deformity results from abnormal changes in mandibular size, orientation, shape, and symmetry. Deformities may develop in the body, condyle, ramus, and symphysis of the mandible.

Patterns of mandibular growth vary widely between individuals. Studies [1,2] on the craniofacial bone patterns of patients with mandibular prognathism have demonstrated that genetic inheritance acts as the crucial catalyst of abnormal growth and development of the mandible. In an investigation of the effects of genetics on human facial bone growth, Hunter et al. [1] reported a significant correlation between heredity and the size of the entire facial bone mass. They noted a higher genetic correlation between the father and child than between the mother and child [1]. Other extrinsic factors, such as mouth breathing or abnormal tongue and lip positioning, also affect the growth and development of the mandible [3]. Alhammadi et al. [4] conducted a systematic review of the literature regarding the global distribution of malocclusion traits in permanent dentition; Class I, Class II, and Class III malocclusion accounted for 74.7% (31–97%), 19.56% (2–63%), and 5.93% (1–20%) of all instances of malocclusion, respectively. Moreover, Lew et al. [5] determined that 12.76% of the Chinese population exhibited Angle Class III malocclusion.

The most commonly used treatment procedures for mandibular deformities in orthognathic surgery (OGS) are sagittal split ramus osteotomy (SSRO) and intraoral vertical ramus osteotomy (IVRO). Unlike SSRO, IVRO does not require rigid (miniplate and miniscrew) or nonrigid (wire) fixation between the proximal (condyle) and distal segments. The objective of IVRO is adaptive condylar remodeling to achieve a balanced position. Furthermore, because IVRO has a lower incidence of inferior alveolar nerve injury, which results in lower lip paresthesia [6,7,8], we preferred this approach in the treatment of mandibular prognathism.

TMJ clicking, an abnormal sensation and sound generated during mandibular movement, is a common symptom of temporomandibular disorder (TMD). Although OGS can be used to reposition the condyle, postoperative complications such as aggravated clicking may occur, causing patients to question the efficacy of mandibular setback surgery. The present study recorded whether patients with mandibular prognathism had preexisting TMJ clicking and evaluated postsurgical TMJ clicking outcomes. We aimed to determine the correlation between preexisting TMJ clicking and the amount of setback caused by the IVRO approach. Assessments were conducted presurgery and one year postsurgery.

## 2. Materials and Methods

### 2.1. Study Design and Sample

This study included a total of 60 patients (120 TMJs) with mandibular prognathism. The inclusion criteria included patients (1) without a history of trauma or relevant syndromes, (2) with no active presurgical mandibular growth, and (3) who were suited for undergoing mandibular setback by IVRO.

### 2.2. Study Variables and Methodology of Data Assessment

We collected data on patient characteristics, surgery, and its outcomes, which included surgery duration, intraoperative blood loss, and amount of mandibular setback. On the basis of our clinical experience and existing evidence [9], a larger setback (>8 mm) posed a greater risk of relapse. We established 8 mm as the cutoff for assessing the incidence of postoperative TMJ clicking. Cone beam computerized (three-dimensional) tomography or panoramic radiography (two-dimensional) images were used to exclude TMJ dislocation in the first 24 h after surgery. Preoperative and one-year postoperative TMJ clicking was confirmed by the surgeon.

### 2.3. Statistical Analysis

SPSS 20.0 (IBM, Chicago, IL, USA) was used for the descriptive and inferential statistical analysis. Descriptive analyses were carried out to examine the sample’s background characteristics, blood loss level, operation time, setback amount, and TMJ clicking. Statistical significance was indicated at *p* < 0.05. A repeated measures analysis was performed to investigate the TMJ clicking difference between patients with a setback amount of ≤8 and >8 mm.

## 3. Results

A total of 30 men (60 TMJs) and 30 women (60 TMJs) met the inclusion criteria to participate in the present study. The mean age of men and women was 22.5 and 23 years, respectively (Table 1). A total of 14 and 12 patients had preexisting TMJ clicking on the right and left sides, respectively (Table 2). Preoperative TMJ clicking occurred on 17 (right side, 9; left side, 8) sides in 10 men but on 9 (right side, 5; left side, 4) sides in 6 women. The mean volume of blood loss was 105.8 mL in men and 95.3 mL in women. The mean surgery duration was significantly longer for men (256.8 min) than for women (241 min). However, no statistically significant differences between men and women in age and intraoperative blood loss were noted.

Dental casts were manually produced to ensure postoperative occlusal stability in accordance with presurgical planning. The right and left mandibular setback amounts (Table 2) were 11.1 mm and 10.9 mm in men, respectively. Women had a 10.7 and 10.0 mm setback on the right and left sides, respectively. No significant difference in the setback amount (right and left side) was noted between the sexes.

The mean setback amount for all patients on the right and left sides were 10.9 mm and 10.5 mm, respectively (Table 2). Postoperative panoramic radiography revealed no TMJ dislocation. The data indicated a lower occurrence of TMJ clicking on the right side; four patients exhibited postoperative TMJ clicking on the right side, and eight patients exhibited it on the left. Postoperative TMJ clicking occurred on five sides (right side, 2; left side, 3) in men but on seven sides (right side, 2; left side, 5) in women. The number of sides exhibiting preoperative TMJ clicking was reduced from a total of 18 (right side, 11; left side, 7). Thus, a total of four new sides (right side, 1; left side, 3) exhibited postoperative TMJ clicking in women. The overall incidence of postoperative TMJ clicking was significantly reduced on both sides (*p* = 0.002) and the right side (*p* = 0.003).

The incidence of preoperative TMJ clicking in women (Table 2) was reduced by a total of six sides (right side, 4; left side, 2). However, women exhibited no considerable reduction in the incidence of postoperative TMJ clicking on either side. Preoperative TMJ clicking in men was improved on 12 sides (right side, 7; left side, 5). No incidence of postoperative TMJ clicking was noted in men, and these patients achieved a significant reduction in the incidence of postoperative TMJ clicking on both the right (*p* = 0.006) and left (*p* = 0.023) sides.

In Table 3, preoperative TMJ clicking was detected at 5 in 25 sides (setback amount ≤ 8 mm) and 21 in 95 sides (setback amount > 8 mm). Postoperative TMJ clicking was observed on a total of four sides (disappeared: 3; no change: 2; new incidence: 2) with a mandibular setback amount of ≤8 mm and on a total of eight sides (disappeared: 15; no change: 6; new incidence: 2) with a mandibular setback amount of >8 mm. The percentage of patients with a setback amount ≤ 8 mm who experienced postoperative TMJ clicking was 16% (4/25); the percentage of patients with a setback amount of >8 mm was 8.4% (8/95). Among male patients with a setback amount of ≤8 mm, TMJ clicking was noted on only 1 (left) side; by contrast, TMJ clicking was observed on a total of three sides for female patients (right side, 1; left side, 2). Among male patients with a setback of >8 mm, TMJ clicking was observed on a total of four sides (right side, 2; left side, 2); TMJ clicking was also noted on a total of four sides for women (right side, 1; left side, 3).

A repeated measures analysis revealed no significant difference in TMJ clicking in patients with a setback amount of ≤8 mm; however, a statistically significant difference (reduced TMJ clicking) was noted on the right side in patients with a setback amount of >8 mm. This finding indicates that a higher amount of setback (>8 mm) does not increase the likelihood of postoperative TMJ clicking.

## 4. Discussion

Mandibular movements facilitate daily functions such as speaking and chewing. The TMJ, which articulates the mandible and cranium, is composed of three parts: the squamous part of the temporal bone, the glenoid fossa (disc and synovial capsule), and the mandibular condyle. TMJ movements (rotation and translation) facilitate four major functions and are mediated by the masticatory muscles; these functions include protrusion (lateral pterygoid muscle), retraction (temporalis muscle), elevation (temporalis muscle, masseteric muscle, and medial pterygoid muscle), and depression (gravity) [10]. The extracapsular ligaments that manage masticatory forces and stabilize TMJ function are the sphenomandibular, stylomandibular, and pterygomandibular ligaments [10,11].

TMD is a dysfunction of the complex system involving the TMJ, disc, and muscles. It manifests in symptoms such as myofascial pain and limited mouth opening. The causes of TMD include the following: malocclusion, bruxism, facial deformity, arthritis, head injury, physical strain, psychological stress, and genetics [12,13,14]. Tonin et al. [15] used magnetic resonance imaging to investigate the correlation between age, sex, and the number of TMD diagnoses. The results indicated that women are more likely than men to develop concomitant conditions, the number of which tends to increase with age. As mentioned, TMJ clicking is a common symptom of TMD. This symptom is characterized by a sound resulting from abnormal TMJ movement due to an altered functional relationship between the articular disc and surface. Generally, TMJ clicking is painless and does not impede mastication [12,13]. Patients with TMJ clicking may experience slight tenderness and swelling in the TMJ area. However, this symptom may gradually worsen, become painful and/or limit mouth opening, and finally impede mastication. Therefore, patients with TMJ clicking should seek medical evaluation and treatment.

Skeletal facial deformity can cause malocclusion and alteration of the TMJ structure. Malocclusion may trigger abnormal muscle activity and tension, which then leads to TMD. The outcomes of OGS in patients with TMJ clicking include: a degree of symptom improvement, no symptom improvement, or worsening of symptoms [16]. Thus, the efficacy of OGS in TMD treatment remains disputed. Some studies [17,18,19] have reported that OGS alleviates TMD symptoms, whereas others [20,21,22] have noted that it may worsen symptoms. Magnusson et al. [18] examined the masticatory function of patients with mandibular protrusion or retrusion after SSRO surgery and reported that OGS yielded beneficial effects on aesthetic appearance, dental occlusion, and TMD signs and symptoms. Dujoncquoy et al. [19] stated that patients with preoperative TMJ symptoms reported significantly improved symptoms after SSRO.

Al-Moraissi et al. [16] reviewed the literature to determine whether OGS exerts a beneficial or a deleterious effect on preexisting TMD. They observed that OGS may lead to changes in the condyle–disc relationship in patients with pretreatment internal derangement. Jung et al. [21] studied the effects of OGS on the TMJ. The restoration of the original condylar position in SSRO is difficult despite miniplate or miniscrew rigid fixation. Excessive pressure against the articular disc or unchanged condylar position may generate a new incidence of TMD or worsen preexisting TMD. Postsurgical condylar resorption may be a physiological regulatory bone remodeling phenomenon or a signal of pathological TMJ disorder. Barone et al. [22] systematically reviewed six databases (PubMed, Cochrane Library, Google Scholar, Scopus, LILACS, and Web of Science) to evaluate the incidence of condylar resorption after OGS. They found that condylar resorption is a possible consequence of OGS and has an incidence rate of 1%–31%. An et al. [23] reported that bone resorption occurred more frequently than bone formation through the investigation of condylar remodeling after a mandibular setback in skeletal Class III deformities by SSRO. Bell et al. [17] noted that anterior disk displacement with reduction through the IVRO approach helps improve TMJ function and alleviate TMD symptoms. Therefore, changes in the condyle–disc relationship are critical in the alleviation of TMD symptoms through OGS.

Westermark et al. [24] investigated the effects of OGS in a total of 1,516 patients with TMD. The negative effects included joint pain, pain when chewing, joint clicking, grinding, headache, and morning headache. The proportion of patients with TMD dropped from 43% before surgery to 28% two years after surgery. The researchers concluded that OGS exerts beneficial effects on the signs and symptoms of TMD. Ellis et al. [25] suggested that OGS improves occlusal force and masticatory ability and performance, which can alleviate TMD symptoms. However, Westermark et al. [24] demonstrated that patients with mandibular retrognathia exhibited no improvement compared with those with mandibular prognathism. Wolford et al. [20] indicated that TMD worsens considerably in patients with preexisting TMD who undergo OGS, particularly mandibular advancement surgery.

Kretschmer et al. [26] evaluated the effects of bimaxillary OGS on TMJ symptoms. They found that OGS alleviates TMJ dysfunction by considerably reducing pain and TMJ clicking. However, TMJ disorders may develop even in patients without preoperative symptoms. Westermark et al. [24] reported that the proportion of 1516 patients with preoperative joint clicking (24%; 19% in 558 men and 27% in 958 women) decreased (20%; 17% in men and 22% in women) after OGS. Westermark et al. [24] noted that the proportion of patients with preexisting TMJ clicking (22%) decreased (17%) two years postsurgery in mandibular prognathism (580 patients) after OGS. When IVRO was performed for the mandibular setback, the incidence of TMD decreased from 42 to 22%. Our findings are consistent with previous studies [24,26]. The ratio of preexisting TMJ clicking was reduced from 21.7 to 10% at the one-year postoperative follow-up. In the present study, the improvement rate was 69.2% (18 of 26 sides) in patients with preexisting TMJ clicking; however, no change was observed in the remaining 30.8% (8 of 26 sides) of patients. Furthermore, TMJ clicking occurred on four new sides in women, whereas new-onset TMJ clicking was not detected in men. After surgery, preexisting TMJ clicking was substantially alleviated in men, but no considerable improvement was noted in women.

We further explored the effects of setback amount on the occurrence of postoperative TMJ clicking. Of the aforementioned four new sides that exhibited TMJ clicking, two exhibited a setback amount of ≤8 mm, and two exhibited a setback amount of >8 mm. The postoperative improvement rate was 71.4% (setback amount > 8 mm) and 60% (setback amount ≤ 8 mm) in patients with preexisting TMJ clicking. Furthermore, we determined that a greater mandibular setback (>8 mm) was not correlated with the incidence of postoperative TMJ clicking in either sex. However, OGS in the treatment of TMD is still quite controversial [26,27,28,29,30].

## 5. Conclusions

In the present study, approximately 66.7% of patients with preexisting TMJ clicking exhibited substantial improvement after mandibular setback by IVRO; this was particularly true for men. A greater mandibular setback (>8 mm) is unlikely to be associated with a considerable increase in the risk of postoperative TMJ clicking. Therefore, the efficacy of OGS in TMJ clicking mitigation remains undetermined.

## Figures and Tables

**Table 1 bioengineering-09-00725-t001:** Patient characteristics.

Variables	Total (n = 60)		Female (n = 30)		Male (n = 30)		Intergender
							Comparison
	Mean	SD	Mean	SD	Mean	SD	*p*-Value
Age (yr)	22.8	4.03	23.0	4.30	22.5	3.80	0.627
Blood loss (mL)	100.6	60.35	95.3	63.38	105.8	57.76	0.499
Operation time (min)	248.6	43.16	241.0	39.77	256.8	45.83	0.033 *

n: Number of patients; *: Significant, *p* < 0.05.

**Table 2 bioengineering-09-00725-t002:** Results on total temporomandibular joint clicking (men: 60 sides; women: 60 sides) in a repeated measures test.

	Total	Right	Left		Female			Male	
Variables	(n = 120)	(n = 60)	(n = 60)	Total	Right	Left	Total	Right	Left
Setback (mm, mean ± SD)	10.7 ± 3.07	10.9 ± 2.88	10.5 ± 3.26	10.3 ± 2.64	10.7 ± 2.34	10.0 ± 2.90	11.0 ± 3.44	11.1 ± 3.36	10.9 ± 3.57
TMJ clicking									
Preoperation (n)	26	14	12	9	5	4	17	9	8
Postoperation (n)	12	4	8	7	2	5	5	2	3
disappeared side (n)	18	11	7	6	4	2	12	7	5
same sides (n)	8	3	5	3	1	2	5	2	3
new sides (n)	4	1	3	4	1	3	0	0	0
Repeated measures test (*p* value)	0.002 *	0.003 *	0.2090	0.532	0.184	0.662	<0.001 *	0.006 *	0.023 *

n: Number of sides; TMJ: temporomandibular joint; *: Significant, *p* < 0.05.

**Table 3 bioengineering-09-00725-t003:** The occurrence possibility of postoperative temporomandibular joint clicking in the amount of setback (≤8 mm and >8 mm) in a repeated measures test.

				Female			Male		
Variables	Total	Right	Left	Total	Right	Left	Total	Right	Left
Amount of setback									
≤8 mm (n, preoperation clicking/Total)	5/25	1/10	4/15	1/10	0/2	1/8	4/15	3/8	1/7
≤8 mm (n, postoperation clicking/Total)	4/25	1/10	3/15	3/10	1/2	2/8	1/15	0/8	1/7
Repeated measures test (*p* value)	0.664	1.000	0.582	0.168	0.500	0.351	0.082	0.351	0.172
>8 mm (n, preoperation clicking/Total)	21/95	13/50	8/45	8/50	5/28	3/22	13/45	8/22	5/23
>8 mm (n, postoperation clicking/Total)	8/95	3/50	5/45	4/50	1/28	3/22	4/45	2/22	2/23
Repeated measures test (*p* value)	0.001 *	0.001 *	0.261	0.159	0.043 *	1.000	0.002 *	0.001 *	0.083

n: Number of sides; *: significant, *p* < 0.05; ≤8 mm (postoperation clicking side; disappeared: 3, no change: 2, new incidence: 2); >8 mm (postoperation clicking side; disappeared: 15, no change: 6, new incidence: 2).

## Data Availability

The data used to support the findings of this study are available from the corresponding author upon request.

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
