# Peer review of "Changes in Preexisting Temporomandibular Joint Clicking after Orthognathic Surgery in Patients with Mandibular Prognathism"

_bioengineering, 2022, doi:10.3390/bioengineering9120725_

Round 1

Reviewer 1 Report

Comments are in the pdf

Author Response

Review Report 1

  1. Line 24: Keywords: must be in alphabetical order and “Amount of setback” is not a keyword

Answer: Keywords is changes as alphabetical order in Line 24 and “Amount of setback” is deleted in Line 25

  1. Line 41: The genetic correlation between a father and his child is higher than that between a mother and her child.

Answer:  The sentence is added a reference. They noted a higher genetic correlation between the father and child than between the mother and child [1].

  1. Line 49: Orthognathic surgery (OGS) involves the repositioning of the condyle

Answer:  In the last paragraph of Introduction,  Previous sentence is change to “TMJ clicking, an abnormal sensation and sound generated during mandibular movement, is a common symptom of temporomandibular disorder (TMD). Although OGS can be used to reposition the condyle, postoperative complications such as aggravated clicking may occur, causing patients to question the efficacy of mandibular setback surgery.”

  1. Line 66: no active growth at surgery 

Answer:  “no active growth at surgery” is changed to “ no active presurgical mandibular growth”

  1. Line 71:  a mandibular setback amount of > 8 mm was considered to be a large.

Answer:  This sentence is changed to “a larger setback (>8 mm) posed a greater risk of relapse”

  1. Line 81:  is there any reason why left is lower than right in both the genders? The amounts of right and left mandibular setback were, respectively, 11.1 and 10.9 mm in men and 10.7 and 10.0 mm in women.

Answer:  The sentences are revised at the second paragraph of Result section, Dental casts were manually produced to ensure postoperative occlusal stability in accordance with presurgical planning. The right and left mandibular setback amounts (Table 2) were 11.1 and 10.9 mm in men, respectively. Women had a 10.7 and 10.0 mm setback in the right and left side, respectively. No significant difference in the setback amount (right and left side) was noted between sexes.

  1. Line 90 and 91: Among patients with postoperative TMJ clicking, 4 exhibited it on the right side, whereas 8 exhibited it on the left side.

Answer:  In the third paragraph of Result section, The data indicated a lower occurrence of TMJ clicking on the right side; four patients exhibited postoperative TMJ clicking on the right side and eight patients exhibited it on the left.  

  1. Line 127 Discussion section : The authors have to provide some correlation/context between the present study and the cited published literature  

Answer:   We cited 2 published papers in the Line 170-176 of Discussion section.  “Postsurgical condylar resorption may be a physiological regulatory bone remodeling phenomenon or a signal of pathological TMJ disorder. Barone et al. [22] systematically reviewed six databases (PubMed, Cochrane Library, Google Scholar, Scopus, LILACS, and Web of Science) to evaluate the incidence of condylar resorption after OGS. They found that condylar resorption is a possible consequence of OGS and has an incidence rate of 1%–31%. An et al. [23] reported that bone resorption occurred more frequently than bone formation through the investigation of condylar remodeling after mandibular setback in skeletal Class III deformities by SSRO.”  

  1. Line 143  They suggested that women are more susceptible to developing several concomitant conditions than are men and the number of these conditions increases with age.

Answer:  The sentence is changed to: The results indicated that women are more likely than men to develop concomitant conditions, the number of which tends to increase with age.    

  1. Line 211   The limitation of present study and the IVRO technique? Further perspectives?  

Answer:  We add at the last two sentences at last paragraph of Discussion section:  “Present study has several limitations, including, a smaller number of patients, and the lack of preoperative and postoperative TMJ magnetic resonance imaging (MRI). Future studies must include a greater number of patients, the use of MRIs, a long-term evaluation after postoperative orthodontic treatment, and investigate the correlation between the setback amount and postoperative disc change.”  

  1. Line 213:  The lasso technique effectively helps avoid postoperative TMJ dislocation.

Answer:  This sentence is deleted. 

Reviewer 2 Report

The study is very interesting and it is worth to be published. The study has some limitations such as the total number of patients is not sufficient to draw generalized strong conclusions. Future studies including a greater number of patients are needed. Also, more long-term evaluation is necessary because TMJ  may be worth it even after the entire surgical orthodontic treatment, including postoperative orthodontic treatment.

Please, mention in the manuscript in the discussion or in the conclusions

Author Response

  1. The study is very interesting and it is worth to be published. The study has some limitations such as the total number of patients is not sufficient to draw generalized strong conclusions. Future studies including a greater number of patients are needed. Also, more long-term evaluation is necessary because TMJ may be worth it even after the entire surgical orthodontic treatment, including postoperative orthodontic treatment.

Please, mention in the manuscript in the discussion or in the conclusions

Answer: 

We add two sentences at the last sentences at last paragraph of Discussion section:   

           “Present study has several limitations, including, a smaller number of patients, and the lack of preoperative and postoperative TMJ magnetic resonance imaging (MRI). Future studies must include a greater number of patients, the use of MRIs, a long-term evaluation after postoperative orthodontic treatment, and investigate the correlation between the setback amount and postoperative disc change.”

Reviewer 3 Report

I reviewed the article “Changes of preexisting temporomandibular joint clicking after orthognathic surgery in patients with mandibular prognathism” and I believe that some issues should be improved.

Introduction:

-          The introduction doesn’t focus on the specific topic of the article;

-          The rationale of the study is missing;

-          The aim should be improved with more details: 1) the population included; 2) the outcomes evaluated; the timing of assessment.

Methods should be strongly improved in terms of the following paragraphs: study design, study sample (TMJ clicking is not precise), study variables, methodology of data assessment, statistical analysis.

It’s also important to clarify what is the role of the 2D panoramic radiography to determine any new incidence of TMJ dislocation after surgery.

Results should be improved. A detailed description of the study sample is missing. Each outcome should be correlated with the variables that could influence them. Descrptive and bivariate statistics are probably insufficient.

Discussion:

-          Line 153: “Facial deformity due to skeletal pattern may lead to malocclusion and altered TMJ structure” Can you please clarify this sentence?

-          Line 164-174: I suggest to deeply consider the results of this recent overview about the condylar changes after orthognathic surgery Barone S, Cosentini G, Bennardo F, Antonelli A, Giudice A. Incidence and management of condylar resorption after orthognathic surgery: An overview. Korean J Orthod. 2022 Jan 25;52(1):29-41. doi: 10.4041/kjod.2022.52.1.29.

-          Study limitations are missing.

-          I believe that you should consider with caution the conclusion “The modified IVRO technique used in this study may help ensure a high amount of 211 setback, increase overlapping bone segments, reduce disturbance to the TMJ, and achieve 212 more satisfactory postoperative outcomes.”. I think that the type of the study design, the included study sample, the absence of a control group, and the statistical analysis used can’t support this sentence.

Author Response

Review Report 3

  1. Introduction:

 The introduction doesn’t focus on the specific topic of the article;

 The rationale of the study is missing;

 The aim should be improved with more details: 1) the population included; 2) the outcomes evaluated; the timing of assessment.

Answer: The last paragraph of Introduction section is revised as follows:

TMJ clicking, an abnormal sensation and sound generated during mandibular movement, is a common symptom of temporomandibular disorder (TMD). Although OGS can be used to reposition the condyle, postoperative complications such as aggra-vated clicking may occur, causing patients to question the efficacy of mandibular setback surgery. The present study recorded whether patients with mandibular prognathism had preexisting TMJ clicking and evaluated postsurgical TMJ clicking outcomes. We aimed to determine the correlation between preexisting TMJ clicking and the amount of setback caused by the IVRO approach. Assessments were conducted presurgery and 1 year postsurgery.

  1. Methods should be strongly improved in terms of the following paragraphs: study design, study sample (TMJ clicking is not precise), study variables, methodology of data assessment, statistical analysis.

Answer:  The Methods section is revised and 3 paragraphs are added: 

2.1 Study design and sample, 

2.2 Study variable and methodology of data assessment

2.3 Statistical analysis

  1. It’s also important to clarify what is the role of the 2D panoramic radiography to determine any new incidence of TMJ dislocation after surgery.

Answer: In the 2.2 Study variable and methodology of data assessment

 The sentence is changed to  “Cone beam computerized (three-dimensional) tomography or panoramic radiography (two-dimensional) images were used to exclude TMJ dislocation in the first 24 hours after surgery.”

  1. Results should be improved. A detailed description of the study sample is missing. Each outcome should be correlated with the variables that could influence them. Descrptive and bivariate statistics are probably insufficient.

Answer:   The Results section is revised.   “A total of 30 men (60 TMJs) and 30 women (60 TMJs) met the inclusion criteria to participate in the present study. The mean age of men and women was 22.5 and 23 years, respectively (Table 1). A total of 14 and 12 patients had preexisting TMJ clicking on the right and left sides, respectively (Table 2). Preoperative TMJ clicking occurred on 17 (right side, 9; left side, 8) sides in 10 men but on 9 (right side, 5; left side, 4) sides in 6 women. The mean volume of blood loss was 105.8 mL in men and 95.3 mL in women. The mean surgery duration was significantly longer for men (256.8 min) than women (241 min). However, no statistically significant differences between men and women in age and intraoperative blood loss were noted.”

“Dental casts were manually produced to ensure postoperative occlusal stability in accordance with presurgical planning. The right and left mandibular setback amounts (Table 2) were 11.1 and 10.9 mm in men, respectively. Women had a 10.7 and 10.0 mm setback in the right and left side, respectively. No significant difference in the setback amount (right and left side) was noted between sexes.”

  1. Discussion section

    Line 153: “Facial deformity due to skeletal pattern may lead to malocclusion and altered TMJ structure” Can you please clarify this sentence?

Answer:   In Line 153 of Discussion section:  we revise the sentence:  “Skeletal facial deformity can cause malocclusion and alteration of the TMJ structure.”

  1. Line 164-174: I suggest to deeply consider the results of this recent overview about the condylar changes after orthognathic surgery Barone S, Cosentini G, Bennardo F, Antonelli A, Giudice A. Incidence and management of condylar resorption after orthognathic surgery: An overview. Korean J Orthod. 2022 Jan 25;52(1):29-41. doi: 10.4041/kjod.2022.52.1.29.

Answer:  We cited 2 published papers in the Line 170-176 of Discussion section.

Postsurgical condylar resorption may be a physiological regulatory bone remodeling phenomenon or a signal of pathological TMJ disorder. Barone et al. [22] systematically reviewed six databases (PubMed, Cochrane Library, Google Scholar, Scopus, LILACS, and Web of Science) to evaluate the incidence of condylar resorption after OGS. They found that condylar resorption is a possible consequence of OGS and has an incidence rate of 1%–31%. An et al. [23] reported that bone resorption occurred more frequently than bone formation through the investigation of condylar remodeling after mandibular setback in skeletal Class III deformities by SSRO.

  1. Study limitations are missing.

Answer:  We add at the last two sentences at last paragraph of Discussion section:   

           “Present study has several limitations, including, a smaller number of patients, and the lack of preoperative and postoperative TMJ magnetic resonance imaging (MRI). Future studies must include a greater number of patients, the use of MRIs, a long-term evaluation after postoperative orthodontic treatment, and investigate the correlation between the setback amount and postoperative disc change.”

  1.  I believe that you should consider with caution the conclusion “The modified IVRO technique used in this study may help ensure a high amount of setback, increase overlapping bone segments, reduce disturbance to the TMJ, and achieve more satisfactory postoperative outcomes.”. I think that the type of the study design, the included study sample, the absence of a control group, and the statistical analysis used can’t support this sentence.

Answer:   In the Conclusion is changed to,  “In the present study, approximately 66.7% of patients with preexisting TMJ clicking exhibited substantial improvement after mandibular setback by IVRO; this was particularly true for men. A greater mandibular setback (>8 mm) is unlikely to be associated with a considerable increase in the risk of postoperative TMJ clicking. Therefore, the efficacy of OGS in TMJ clicking mitigation remains undetermined.”

Round 2

Reviewer 1 Report

Revisions are to my satisfaction

Reviewer 3 Report

The manuscript was adequately improved.